# BOIL: Learning Environment Personalized Information

## Abstract

Navigating complex environments poses challenges for multi-agent systems, requiring efficient extraction of insights from limited information. In this paper, we introduce the Blackbox Oracle Information Learning (BOIL) process, a scalable solution for extracting valuable insights from the environment structure. Leveraging the Pagerank algorithm and common information maximization, BOIL facilitates the extraction of information to guide long-term agent behavior applicable to problems such as coverage, patrolling, and stochastic reachability. Through experiments, we demonstrate the efficacy of BOIL in generating strategy distributions conducive to improved performance over extended time horizons, surpassing heuristic approaches in complex environments.

## 1 Introduction

Since the 1980s, the concept of employing multiple interacting agents to accomplish various tasks has garnered significant attention in the fields of robotics, artificial intelligence, and distributed systems. Previous research endeavors have demonstrated successful methods for utilizing multiple robots to tackle diverse challenges such as area coverage, patrolling, and reachability [1] Bierkens (2016); Clempner (2018); Díaz-García et al. (2023); Hespanha et al. (1999); Langley et al. (2021); Rahili et al. (2017); Stern et al. (2006). These efforts have embraced a multitude of approaches spanning game theory, genetic algorithms, greedy heuristics, and more, reflecting the multifaceted nature of the problems at hand.

Despite this breadth of exploration, a fundamental trade-off persists between achieving optimal solutions, maintaining computational tractability, and ensuring scalability concerning the number of agents or the size of the environment. Some methodologies adopt an independence assumption among agents to facilitate tractability for a large number of participants Díaz-García et al. (2023); Langley et al. (2021), thus sacrificing potential optimality for scalability. The main objective of this paper is fine-grained control of these trade-offs.

In this paper, we operate under the assumption of having access to an oracle whose information is indirectly accessible, and our aim is to devise a computationally scalable approach to extract insights from this oracle. Our focus lies in demonstrating the feasibility of extracting information from an oracle whose behavior adapts to environmental changes. Leveraging the Pagerank algorithm Page et al. (1998), renowned for its computational efficiency, we propose a scalable method for extracting information from the oracle. Crucially, our approach remains independent of the number of agents involved, and we formulate the problem as a common information maximization [2] task.

We begin by providing a comprehensive review of existing literature concerning the aforementioned tasks, followed by an introduction to Non-reversible Markov chains and Supervised PageRank techniques. Subsequently, we illustrate the construction of our proposed process within the context of the coverage problem, termed BOIL (Blackbox Oracle Information Learning), emphasizing its ability to distill information from the oracle into learnable parameters. Many real world problems like mobile sensor coverage Rahili et al. (2017), forest fire detection Alsammak et al. (2022), agricultural monitoring Albani et al. (2017), traffic data collection Elloumi et al. (2018), etc. can be modeled

---

[1] Reachability can be seen as the task which creates a path for agents that allows fast reaction when an intruder is detected.

[2] There are multiple ways to define common information. We use the definition given by Liu et al. (2010).

using coverage. With the foundational structure of BOIL established, we present results that demonstrate its efficacy, particularly in scenarios where a higher number of parameters is computationally feasible. Furthermore, we discuss how our method can be used to address patrolling and reachability tasks.

The primary contribution lies in showcasing the utility of extracted information from a blackbox oracle, offering theoretical insights and practical implications for enhancing the performance and robustness of multi-agent systems in complex environments. We run simulated experiments to support the claims about the effectiveness of the process.

## 2    RELATED WORKS

Several approaches have been proposed in the literature to address the challenges of coverage, patrolling, and reachability in multi-agent systems. Stern et al. (2006) introduced a genetic algorithm-based technique for coverage, optimizing a complex fitness function dependent on factors such as distance, travel time, and visibility within a discretized space. While providing complete trajectories, this approach becomes computationally intractable for large environments or a high number of agents due to its dependency on the number of agents. Rahili et al. (2017) presented a game-theoretic framework for distributed coverage of mobile sensors, aiming to find equilibrium positions for sensors using reinforcement learning to converge to Nash equilibrium. This approach incorporates utility functions for agents directly into transition probabilities, assuming reversible agent movements [3] and focusing on reaching equilibrium via agent interactions. Our work generalizes to non-reversible agent movements. In both works, the planning and control of agents is tightly coupled making it hard to tweak individual components separately. In this work, we provide a way to loosely couple planning and control allowing a fine-grained ability to deal with various trade-offs.

Mathew & Mezić (2011) showed that it is possible create control inputs for a group of agents based on a probabilistic high level plan. Leveraging Ergodic control, they addressed the coverage problem with a given distribution for space coverage, defining first and second-order dynamics of agents. Their work shows that we can create a loose coupling between planning and control, the main focus is on control. In contrast, our work focuses on the planning part of the problem. Abraham & Murphey (2018) demonstrated a decentralized ergodic control method for coverage, focusing on achieving a given target distribution. They claim that their work can be extended to pursuit-evasion games but do not a give mathematical formulation. Stochastic reachability, akin to the work of Hespanha et al. (1999), pertains to solving agent trajectories to maximize the probability of reaching specific locations within a fixed time. We discuss how it is possible to give a mathematical formulation for stochastic reachability with minor changes in the original formulation.

Patrolling tasks present distinct challenges, where the number of objective points may be significantly fewer than the total environment size. Game-theoretic approaches, such as Stackelberg games Clempner (2018); Gan et al. (2018) and Nash equilibrium strategies Langley et al. (2021), have been widely adopted. Additionally, constraint-based methods Díaz-García et al. (2023) have been explored, offering scalable solutions for multi-agent patrolling.

## 3    PRELIMINARIES

### 3.1    NON-REVERSIBLE MARKOV CHAIN

Consider a Markov chain with a finite or countable state space $S$, characterized by transition probabilities $P(u \rightarrow v)$ for $u, v \in S$. To ensure a valid probability distribution, the following condition must hold:

$$\sum_{v \in S} P(u \rightarrow v) = 1 \ \forall u \in S \tag{1}$$

**Definition 1.** *Let $\pi$ be a probability distribution on the states $S$, such that $\sum_{u \in S} \pi(u) = 1$. A distribution is termed the stationary distribution of the Markov chain if the following condition*

---

[3]Reversible movements implies that if an agent can take action $a$ to transition from state $s$ to $s'$, then there must exist an action $a'$ such that the agent can transition from $s'$ back to $s$.

*holds:*

$$\sum_{u \in S} \pi(u)P(u \to v) = \pi(v) \quad \forall u, v \in S \tag{2}$$

The equation referenced as 2 defines the global balanced condition. It is worth noting that this condition is weaker than the well-known detailed balanced condition presented below in equation 3:

$$\pi(u)P(u \to v) = \pi(v)P(v \to u) \; \forall u, v \in S \tag{3}$$

When the above condition holds, the Markov chain is considered reversible. Metropolis et al. (1953) and Hastings (1970) introduce Metropolis-Hastings sampling (MH), a technique for sampling from reversible Markov chains with convergence guarantees. However, extending it to non-reversible Markov chains is non-trivial. Bierkens (2016) demonstrates the construction of a technique analogous to MH for the non-reversible case.

Bierkens (2016) defines the non-reversible Hastings ratio as:

$$R_\Gamma(u, v) := \begin{cases} \frac{\Gamma(u,v) + \pi(v)Q(v,u)}{\pi(u)Q(u,v)} & \pi(u)Q(u,v) \neq 0 \\ 1 & \text{otherwise} \end{cases} \tag{4}$$

where $\Gamma(u, v) := \pi(u)P(u \to v) - \pi(v)P(v \to u)$. However, to ensure non-negative values of $\Gamma(u, v)$, the following constraint must hold:

$$-\pi(v)Q(v, u) \leq \Gamma(u, v) \leq \pi(u)Q(u, v) \quad \forall u, v \in S \tag{5}$$

## 3.2 SUPERVISED PAGERANK

Page et al. (1998) defines PageRank as a method for ranking the importance of different web pages. At its core, the movement of a user on the World Wide Web is modeled as a Markov chain where the state space is defined as the set of web pages. Vanilla PageRank assumes that a user will move uniformly randomly from the current page to all pages the user can directly access. Let $G(V, E)$ be a graph where the vertices represent the web pages and the edges represent the links that the user can use to move from one page to another. Then, PageRank can be mathematically defined as Page et al. (1998):

**Definition 2.** *Let $\pi$ be a probability distribution on the vertex set $V$, i.e., $\sum_{v \in V} \pi(v) = 1$. We need to find $\pi$ such that $c$ is maximized with the constraints given below:*

$$\pi(v) = \alpha(v) + c \sum_{\{u | (u,v) \in E\}} \frac{\pi(u)}{|\{w | (u, w) \in E\}|} \quad \forall v \in V \quad \text{\textcolor{blue}{$\alpha$ is a vector over the web pages.}} \tag{6}$$

The vector $\alpha$ serves two purposes: 1) It acts as a source for the ranking process, and 2) it ensures that the Markov chain is strongly connected, thus irreducible.

The above approach can be extended to generate a user-specific ranking of the nodes by adjusting the transition probabilities of the Markov chain. Zhukovskiy et al. (2014) introduces a method that formulates the process of learning personalized PageRank as an optimization problem. This method models the transition probabilities and node vectors using functions $f : X \times V \to \mathbb{R}$ and $g : Y \times E \to \mathbb{R}$, where $X$ and $Y$ represent sets of parameters. We leverage the extension of this technique for learning Supervised PageRank with gradient-free optimization methods and approximate oracles Bogolubsky et al. (2016).

## 4 BLACKBOX ORACLE INFORMATION LEARNING (BOIL)

In this section, we will begin by formulating the problem environment. Subsequently, we will demonstrate how information is extracted from the environment structure to address the coverage problem utilizing movement constraints (Flow constraints) enforced by the environment. In the coverage task, we consider a node is covered if it comes in visibility of any agent, and the agents want

to cover all nodes equally as many times as possible.[4]. It is important to note that throughout this work, we assume homogeneity among agents, meaning they possess identical physical capabilities unless stated otherwise. Proofs for all Theorems are deferred to the Appendix.

## 4.1 COVERAGE PROBLEM FORMULATION

Similar to prior research, we represent the environment with a graph. However, a key distinction lies in our utilization of an undirected graph $\mathcal{G}(V, E)$ to encode the topography or environment semantics alongside a directed graph $\mathcal{G}_d(V_d, E_d)$ to model the potential movement space of the robots/agents. Such a formulation allows a rich representation for real world settings like cases when agents can go down a hill but cannot come up the same way because of physical constraints or model a trapdoor like situation that arise in case of fire exits and one-ways in urban settings. It's important to note that all vertices $v \in V_d$ possess self-loops, denoted by $(v, v) \in E_d \; \forall \, v \in V_d$, indicating that $\mathcal{G}_d$ is not a simple directed graph[5]. Additionally, we assume that $\mathcal{G}_d$ is strongly connected. This assumption is made to prevent scenarios where the agent reaches states from which it cannot return to its initial state or where certain parts of the graph are unreachable regardless of the number of steps taken. Furthermore, we assume that the traversal time for any edge in $E_d$ is independent of the agent's previous actions.

Stern et al. (2006) utilizes a visibility matrix as a map listing nodes visible from a particular node. This matrix formally defines a fitness function for the genetic algorithm in prior research. Here, we define visibility as a function $V_s : E_d \rightarrow [0, 1]^{|V|}$, where $V_s((u, v))(w)$ represents the value for node $w \in V$ when $V_s$ is assessed on $(u, v) \in E_d$ which allows us to define unidirectional vision for agents. This value gives the probability that node $w$ is visible when an agent crosses edge $(u, v)$.

**Definition 3.** *Let $V_s((\bar{u}, \ldots, \bar{v}))$ denote the visibility for a path $(\bar{u}, w_1, w_2, \ldots, w_{l-1}, \bar{v})$, where $T_{\bar{u}\bar{v}}$ denotes the time required for the agent to traverse the path. Let $\mathcal{V}_w(t) : [0, T_{\bar{u}\bar{v}}] \rightarrow \{0, 1\}$ be an indicator function, which is 1 if the node $w$ is visible at time $t$. Then,*

$$\forall w \in V, \; V_s((\bar{u}, \ldots, \bar{v}))(w) = \frac{1}{T_{\bar{u}\bar{v}}} \int_0^{T_{\bar{u}\bar{v}}} \mathcal{V}_w(t) dt$$

Consider a system with $n$ agents. Let the function $h^i_{(u,v)}(t)$ is 1 for all $t$ when agent $i$ is crossing the edge $(u, v)$, and otherwise 0. Essentially, these functions are generated from an oracle that provides paths for the agents. Suppose we make a vector $Y_t$, which represents the probability that a node in $V$ is visible at time $t$ from any agent. Then we can write $Y_t$ as following:

$$Y_t = \mathbf{1} - \prod_{i=1}^{n} \left( \mathbf{1} - \sum_{(u,v) \in E_d} h^i_{(u,v)}(t) V_s((u, v)) \right) \tag{7}$$

where the product of vectors represents the element-wise multiplication. For the rest of the work, consider that a direct vector of products represents the element-wise product unless otherwise mentioned. We can define the node visibility probability for the time duration $[0, T]$ as $P_V(w) = \left( \int_0^T Y_t(w) dt \right) / T$. The coverage problem can now be defined as maximizing the common information Liu et al. (2010) for random variables that represent the node visibility. We use Theorem 1 given below to find a bound on the integral and reduce the problem to minimizing the loss function $\mathcal{L}$ as follows:

$$\mathcal{L} = \sum_{w \in V} -\mathcal{A}(w) \log \mathcal{A}(w) \quad \text{where} \quad \mathcal{A}(w) = \sum_{(u,v) \in E_d} P((u, v)) V_s((u, v))(w) \tag{8}$$

The details of the reduction are deferred to the appendix along with proof of Theorem 1. The proof also covers why the loss is independent of the agent count.

---

[4]In this work, we make a distinction between patrolling the area and patrolling specific areas in the environment. We call the first one coverage and the latter patrolling.

[5]Directed graphs without self-loops are referred to as simple directed graphs.

**Theorem 1.** *Suppose we define,*

$$P((u,v)) = \frac{1}{nT} \sum_{i=1}^{n} \int_0^T h^i_{(u,v)}(t)dt \quad and \quad \mathcal{X}^i(t) = \sum_{(u,v) \in E_d} V_s((u,v))(w)h^i_{(u,v)}(t)$$

*then $P$ is a probability distribution on the edges $E_d$ and the following holds:*

$$nT \sum_{(u,v) \in E_d} P((u,v))V_s((u,v))(w) \geq \int_0^T Y_t(w)dt \ \forall w \in V$$

$$T \sum_{(u,v) \in E_d} P((u,v))V_s((u,v))(w) \leq \int_0^T Y_t(w)dt \ \forall w \in V$$

(9)

*Furthermore,*

$$nT \sum_{(u,v) \in E_d} P((u,v))V_s((u,v))(w) \leq \sum_{1 \leq i < j \leq n} \int_0^T \mathcal{X}^i(t)\mathcal{X}^j(t)dt + \int_0^T Y_t(w)dt$$

## 4.2 SOLVING AS FLOW CONSTRAINT PROBLEM

Any agent can only cross the edge $(u,v) \in E_d$ only if the agent is on the node $u \in V_d$. Moreover, the agent can stay at a node $u$ only if it has reached node $u$. All the agents in the system are constrained by the structure of the directed graph $\mathcal{G}_d$. Following the above argument, we can decompose $P((u,v))$ into two probabilities (shown in Theorem 2): 1) Probability that the agent is on node $u$ represented as $\pi(u)$ and 2) probability that the agent moves along the edge $(u,v)$ given that the agent is on the node $u$ represented as $P(u \to v)$. Hence,

$$\implies P((u,v)) = \pi(u)P(u \to v) \quad and \quad \sum_{\{v|(u,v) \in E_d\}} P(u \to v) = 1 \tag{10}$$

**Theorem 2.** *The function $\pi : V_d \to \mathbb{R}$ is a probability distribution on the nodes in $V_d$.*

Furthermore, the net flow of the probability should be balanced across the graph, implying that $P(u \to v)$ for all $(u,v) \in E_d$ and $\pi(u)$ for all $u \in V_d$ have values such that the global balanced condition (Equation 2) is satisfied. Hence, the movement can be modeled as a Markov chain with the nodes $V_d$ as the state space. However, we do not know the transition probabilities $P(u \to v)$.

Previous works have studied Mathew & Mezić (2011); Miller et al. (2015); Abraham & Murphey (2018) how multiple agents can approximate a given distribution defined similarly to the definition of $P((u,v))$ as given in theorem 1. Please observe that the loss function 8 only depends on the transition probabilities $P((u,v))$, implying that we need to only find the correct values for $P(u \to v)$ that minimizes the loss function. Hence, we can model the problem as a Supervised PageRank 3.2 optimization problem. Building on the work of Bogolubsky et al. (2016), we get Algorithm 1.

Note how we have only constrained the flow of probability as Pagerank only ensures the global balanced condition 2. The oracle is supposed to give continuous paths but we solved for only the softer probabilistic constraint. In essence, running Algorithm 1 allows us to access some information from the oracle. While previous work does not require an oracle, they use hard constraints, thereby restricting custom control over the design trade-offs. Using the oracle formulation allows us to do *fine-grained estimation*.

## 4.3 FINE GRAINED ESTIMATION

The estimator that we used in the coverage problem uses a distribution over the node space. However, the parameters required to solve for the node space might be below optimal for the available compute. Furthermore, increasing the resolution of the graph in the node space can lead to an increase in the parameter space larger than the available compute. Moreover, it might be favorable to get additional information about the movement of the agents along the time axis in place of spatial information.

---

### Algorithm 1: BOIL Algorithm

---

**Require:** $\mu > 0$ step size, $N \geq 1$ number of steps, $\boldsymbol{p}_0$ vector of $P(u \rightarrow v)$ $\forall (u, v) \in E_d$, Graph $\mathcal{G}_d$, Loss Function $\mathcal{L}$

    **Begin Procedure**

    $k \leftarrow 0, m \leftarrow |\boldsymbol{p}_0|, \boldsymbol{x} \leftarrow PAGERANK(\boldsymbol{p}_0, \mathcal{G}_d)$          ▷ Do pagerank to get values of $\pi(u)$

    **while** $k < N$ **do**

        $\boldsymbol{r} \leftarrow$ Random vector on $m$ dimensional sphere

        $\boldsymbol{q} \leftarrow$ Normalize values of $(\boldsymbol{p}_k + \boldsymbol{r})$ to ensure $\sum_{\{v | (u,v) \in E_d\}} P(u \rightarrow v) = 1$

        $\boldsymbol{y} \leftarrow PAGERANK(\boldsymbol{p}_k + \boldsymbol{r}, \mathcal{G}_d)$

        $\boldsymbol{g} \leftarrow m(\mathcal{L}(\boldsymbol{p}_k + \boldsymbol{r}, \boldsymbol{y}) - \mathcal{L}(\boldsymbol{p}_k, \boldsymbol{x}_k))\boldsymbol{r}$          ▷ Gradient for update

        $\boldsymbol{p}_{k+1} \leftarrow$ Normalize values of $(\boldsymbol{p} - \mu\boldsymbol{g})$ to ensure $\sum_{\{v | (u,v) \in E_d\}} P(u \rightarrow v) = 1$

        $\boldsymbol{x} \leftarrow PAGERANK(\boldsymbol{p}_{k+1}, \mathcal{G}_d)$

        $k \leftarrow k + 1$

    **end while**

    $k \leftarrow \arg\min_k \{\mathcal{L}(\boldsymbol{p}_k, \boldsymbol{x}_k)) : k \in \{0, \ldots, N-1\}\}$

    **Output:** $\boldsymbol{p}_k, \boldsymbol{x}_k$

    **End Procedure**

---

Observe how the presented technique only requires a strongly connected state space, and the visibility function depends only on the state and not on the time the state is reached. Now, we show various methods to increase the state space, which gives more information about the system with a relatively low increase in the parameter space.

Consider the coverage problem itself. Suppose we are given a continuous path $(\bar{u}, w_1, w_2, \ldots, w_{l-1}, \bar{v})$ and we want to find out the probability the agent should take this path to minimize the loss.

**Theorem 3.** *Given a path* $(\bar{u}, w_1, w_2, \ldots, w_{l-1}, \bar{v})$*, the edge set* $E_d$ *can be modified to add an additional edge between the node* $\bar{u}$ *and* $\bar{v}$ *represented as* $(\bar{u}, \ldots, \bar{v})$*. Optimizing the loss function* $\mathcal{L}$ *on the original edge set* $E_d$ *is the same as optimizing the loss function over the modified edge set. Furthermore, the probability that the agent takes the path is the probability for the edge* $(\bar{u}, \ldots, \bar{v})$ *found by optimizing the loss on the modified edge set.*

Observe that Theorem 3 can be applied repeatedly, and it increases only 1 parameter per path. Note how the base formulation can be obtained by repeatedly applying Theorem 3 on the empty edge set with paths of unit length. Adding higher length paths puts a hard continuity constraint in addition to the soft probabilistic one thus giving a fine-grained control. Suppose we can handle twice the number of parameters and want to get more information about the movement of the agents in the time domain, then we can use Theorem 4.. Let $\mathcal{P}(p)$ represent the set of all countable partitions of the interval $[0, T]$ for $p \in (0, 1)$ such that:

$$\int_0^T I(t)\, dt = pT, \qquad \forall\, (t_1, t_2, \ldots) \in \mathcal{P}(p) \quad \text{where} \quad I(t) = \begin{cases} 1 & t \in t_i, \text{i is odd} \\ 0 & \text{otherwise} \end{cases} \tag{11}$$

For any partition $(t_1, t_2, \ldots) \in \mathcal{P}(p)$, we can split the $h^i_{(u,v)}(t)$'s into two functions $\hat{h}^i_{(u,v)}(t)$ and $\bar{h}^i_{(u,v)}(t)$ such that:

$$\hat{h}^i_{(u,v)}(t) = \begin{cases} h^i_{(u,v)}(t) & t \in t_i, \text{i is odd} \\ 0 & \text{otherwise} \end{cases} \qquad \bar{h}^i_{(u,v)}(t) = \begin{cases} h^i_{(u,v)}(t) & t \in t_i, \text{i is even} \\ 0 & \text{otherwise} \end{cases} \tag{12}$$

Observe that $h^i_{(u,v)}(t) = \hat{h}^i_{(u,v)}(t) + \bar{h}^i_{(u,v)}(t)$.

**Theorem 4.** *For any partition* $(t_1, t_2, \ldots) \in \mathcal{P}(p)$*, let*

$$\hat{P}((u,v)) = \frac{1}{npT} \sum_{i=1}^n \int_0^T \hat{h}^i_{(u,v)}(t)dt \quad and \quad \bar{P}((u,v)) = \frac{1}{n(1-p)T} \sum_{i=1}^n \int_0^T \bar{h}^i_{(u,v)}(t)dt$$

*Then, $\hat{P}$ and $\bar{P}$ form a probability distribution over the edges $E_d$. Furthermore, when decomposed in edge transition probabilities, represented as $\hat{P}(u \to v)$ and $\bar{P}(u \to v)$ for edge $(u,v) \in E_d$, and a distribution over the nodes, represented as $\bar{\pi}(u)$ and $\hat{\pi}(u)$ for vertex $u \in V_d$, respectively, then the following holds:*

$$\pi(u) = p\hat{\pi}(u) + (1-p)\bar{\pi}(u)$$

*Furthermore, the following constraints are sufficient to ensure flow constraints:*

$$\sum_{\{v|(u,v)\in E_d\}} \hat{\pi}(u)\hat{P}(u \to v) = \hat{\pi}(v) \quad and \quad \sum_{\{v|(u,v)\in E_d\}} \bar{\pi}(u)\bar{P}(u \to v) = \bar{\pi}(v)$$

To ensure that the found distributions are as different as possible, we can optimize for the following loss instead of the original loss $\mathcal{L}$:

$$\mathcal{L}_\Lambda = \mathcal{L} - \frac{1}{2}\sum_{(u,v)\in E_d} \Lambda_{(u,v)}\left(\hat{P}((u,v)) - \bar{P}((u,v))\right)^2 \tag{13}$$

where $\Lambda : E_d \to \mathbb{R}$ is some fixed function.

As the theorem holds for any partition, optimizing the loss will give us the distributions that minimize the loss over all partitions in $\mathcal{P}$.

## 5 EXPERIMENTS & RESULTS

We will analyze the results obtained on a relatively complex environment while we defer the experiments related to scalability to the Appendix C. All computation was done using only 19 CPU cores without any GPU acceleration.

Our experiments model agents with unidirectional visibility and irreversible movement which is a limitation for many previous works even though both conditions are found ubiquitously in real world situations. Works that consider general agent movement and visibility become intractable for the size of environment and agent count that we consider. To our knowledge, no baseline provides a fair comparison for evaluation. Hence we select a few strategies that fit the requirements of the task.

### 5.1 AGENT STRATEGIES

**Random Agent**. As the name suggests, the agents do a random walk. The agent chooses an edge to move along from the current position with a uniform probability.

**OptRandom Agent**. The difference here is that the agent will randomly sample any edge from the set $E_d$ uniformly and traverse it. It is possible that the agent teleports to node $w$ after traversing edge $(u,v)$ because the next sampled edge is $(w,x)$. We add this strategy to understand the distribution the agent will reach when flow constraints are not followed[6]. As it is an unconstrained solution, it also serves as a strong baseline for comparison.

**Frontier Agent**. Frontier agent prioritizes the unexplored regions. Originally, it is an exploration algorithm Yamauchi (1997; 1998), and hence, we modify it for the coverage problem. The vanilla frontier exploration technique is deterministic[7], however, the agent can get stuck at an optima after exploring the entire region. Hence, we create a randomized version that utilizes a count vector $C$. $C(w)$ gives the number of times node $w$ was visible to the agent. We define the transition probability as:

$$P(u \to v) \propto \frac{1}{|V((u,v))|}\sum_{w\in V((u,v))} \frac{1}{\max(C(w), 10^{-6})} \tag{14}$$

where $w \in V((u,v))$ is a shorthand for iterating over the non-zero entries of the vector. The probability is normalized and used for transition. The function motivates to cover areas uniformly.

---

[6]The flow constraint is an additional constraint on the system. So, it is possible that the agent which can teleport can do better, which we will see quantitatively later.

[7]The algorithms are largely deterministic. There can be points where the algorithm gives some actions the same score in which case it can choose randomly in those actions.

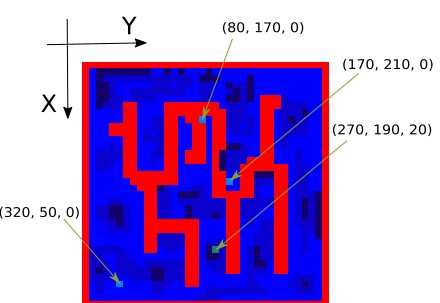

Figure 1: The figure depicts the small environment with a color encoding easier for understanding the topology. The *cyan-like* colors are pixels that encode the elevation, and that extra data is supposed to be stored for nodes representing those pixels.

**Sample Agent**. The sample agent uses the probability distribution we found using BOIL over the edges $E_d$ to sample its path. We use the technique shown in subsection 3.1 for doing MH sampling. However, we cannot sample directly as the proposition distribution needs to be a function of the current position so that we can get a continuous path. This slows down the mixing process, but we still have the guarantee that the distribution will mix. To effectively cover the area, we use a Q-function:

$$Q(u,v) = 1 + \frac{\lambda}{|V((u,v))|} \sum_{w \in V((u,v))} \frac{1}{\max(C(w), 10^{-6})} \tag{15}$$

where $C(w)$ is the count vector similar to the Frontier agent. The parameter $\lambda$ controls its preference towards frontier exploration. We use $\lambda = 10$ for all experiments.

**Comm Frontier Agent**. It is similar to the frontier agent except that all the agents maintain a single $C$ vector. Furthermore, the probability is scaled by the number of agents.

**Comm Sample Agent**. It is similar to the Sample agent except that all the agents maintain a single $C$ vector. The number of agents scales the frontier part of Q-function.

**Optimal Agent**. The optimal agent is similar to the OptRandom agent. The agent does not care about the continuity of the path and samples directly from the distribution obtained over $E_d$ using MH for non-reversible Markov chains. The idea is to show the long horizon behavior for Sample agents and Sample Comm agents in simulation for analysis purposes.

## 5.2 EXPERIMENTS

The small environment comprises a $36 \times 36$ field, featuring tall walls and uneven topography illustrated in Figure 1. In the depiction, red blocks denote tall walls, impassable and invisible to the agent. Elevation levels are represented by varying shades of blue, with lighter shades indicating lower elevation and darker shades signifying higher elevation. We run simulation with homogeneous teams with 8 agents with unidirectional visibility.

The agent's movement is constrained, permitted only from lighter to darker shades in sequential order. However, it can descend directly from a darker shade to any blue-colored pixel. Elevational differences significantly impact visibility, with higher elevations offering greater visibility and vice versa. The *cyan-like* colored pixels convey an elevation in the same three levels just that we will use these points specifically to understand the behaviors of the different agents.

We conducted simulations ten times, each spanning $10^5$ steps, to explore the variance resulting from randomness. Figure 2 presents quantitative visibility counts, while Figure 3 displays the total variation distance (Definition 4 Gibbs & Su (2002)) of empirical distributions to those obtained through the BOIL process.

**Definition 4.** *Given two probability $\mu$ and $\nu$ on the discrete state space $S$,*

$$d(\mu, \nu) = \frac{1}{2} \sum_{x \in S} |\mu(x) - \nu(x)|$$

*where $d$ is the total variation distance Gibbs & Su (2002).*

In the context of the uniform coverage problem, the objective is to ensure that agents explore every area equally. Figure 2 reveals that Optimal Agents exhibit a prominent peak, indicating effective

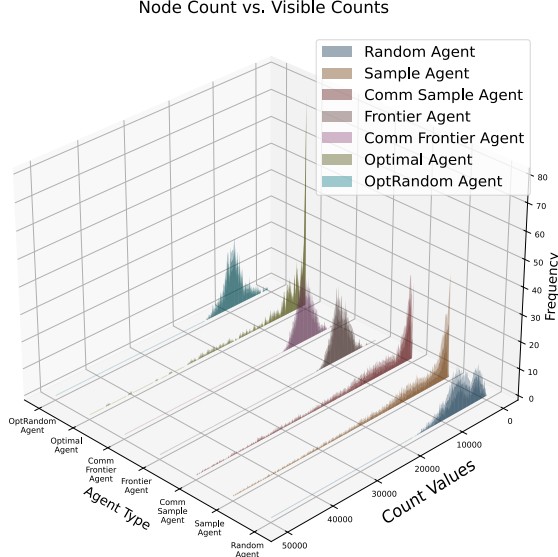

Figure 2: The Z-axis illustrates the counts of visible nodes corresponding to values on the Y-axis, while the X-axis denotes the types of homogeneous agents considered in the experiment. Curves' opacity signifies variations in values, with lower opacity indicating the upper end of the variance and higher opacity representing the lower end for frequencies. The central divider indicates the mean frequencies. **BOIL Distribution Strategies:** *Sample Agent, Comm Sample Agent, Optimal Agent*

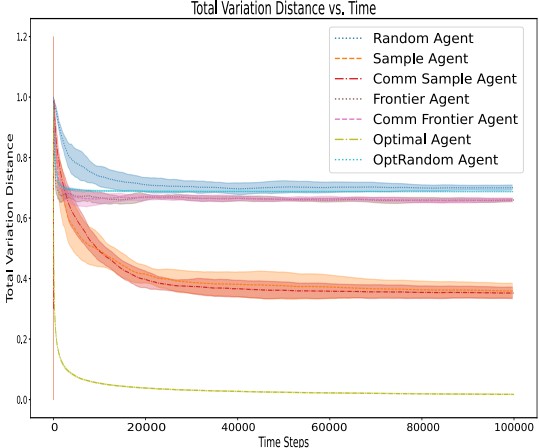

Figure 3: The figure depicts Total variation distance between the approximate distribution of the agent trajectories and the distribution found using the BOIL process for the small environment for all the timesteps. **BOIL Distribution Strategies:** *Sample Agent, Comm Sample Agent, Optimal Agent*

utilization of visibility across different locations. Interestingly, all strategies except for Sample and Comm Sample agents demonstrate distributions similar to OptRandom agents, suggesting suboptimal coverage patterns.

Despite their similarities to Optimal agents, Sample and Comm Sample strategies fail to achieve the optimal distribution in $10^5$ steps, as evident from Figure 3. While the distance of the Optimal agent converges to zero, it plateaus for the Sample and Comm Sample agents, though it continues to decrease slowly. Furthermore, the figure illustrates significantly higher and similar distances for all other strategies.

These observations highlight the limitations of frontier exploration techniques, particularly in complex environments and over extended time horizons. Although Sample-based strategies exhibit similar distributions to Optimal agents, they fail to achieve the desired optimal distribution.

Figure 4 provides a cumulative count of how many times a particular point was visible. It is intriguing to observe that the Optimal agents exhibit a pronounced preference for observing the corner point significantly more times than others. Conversely, both Frontier and Comm Frontier agents exhibit higher counts than Sample and Comm Sample agents in 3 out of 4 points. Notably, the point with a higher elevation, visible more frequently from Sample and Comm Sample agents, is surrounded by areas at higher elevations. This observation suggests that the simple sampling-based strategy effectively leverages the distribution obtained from BOIL to prioritize high-visibility points.

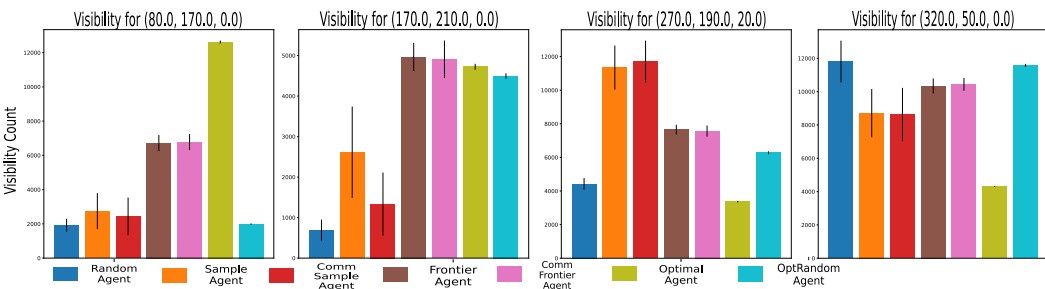

Figure 4: The figure shows the visibility counts for the *cyan-like* colored pixels in figure 1 for the different agent strategies. The bars shows the mean and the errorbars show the variance.

# 6 DISCUSSION

## 6.1 EXTENSION TO PATROLLING AND REACHABILITY

The technique can be extended to other problems like patrolling and reachability problems. Patrolling can be defined as the problem where the agents want to patrol certain points on the graph at all times. Reachability can be defined as the problem where the agent wants to ensure that it can reach any particular place in a certain amount of time or not.

Consider a set of vertices $V_p \subseteq V$ that needs to be patrolled by the agents for the patrolling problem. Keeping the same setting as in the case of coverage problem discussed before, we only need to modify the loss function to the following:

$$\mathcal{L}_p = \sum_{w \in V_p} -\mathcal{A}(w) \log \mathcal{A}(w) \quad \text{where} \quad \mathcal{A}(w) = \sum_{(u,v) \in E_d} P((u,v)) V_s((u,v))(w) \quad (16)$$

The modified loss only minimizes the expected information from points of interest and discards the information from other nodes. For the reachability problem, in place of a visibility function, we define a reachability function $R : E_d \rightarrow (0,1)^{|V|}$ where $R((u,v))(w)$ represents the probability that an agent can reach the node $w \in V$ if the agent is traversing the edge $(u,v)$ within a predefined time $T_R(w)$. We can now replace the visibility function in the formulation for the coverage problem.

## 6.2 CONCLUSION & FUTURE WORK

In this paper, we introduced the Blackbox Oracle Information Learning (BOIL) process as a scalable solution for extracting valuable insights from the environment structure in multi-agent systems. Leveraging the Pagerank algorithm and information theory, BOIL enables the extraction of information about long-term agent behavior. We demonstrated the flexibility of the formulation by applying it to various problems such as coverage, patrolling, and stochastic reachability, all converted into common information maximization problems solvable using the same technique.

An important assumption of our work is the availability of reliable information about the environment. With the entire process being offline, it is not possible to deal with dynamic changes in the environment. To address this limitation, future work aims to extend the framework to allow for online updates in the BOIL process, coupled with a controller capable of independently utilizing extracted information. This direction holds promise, as BOIL is an iterative process seemingly independent of noise in control policies.

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

## A BOIL

**Lemma 1.** *Given two continuous paths $(\bar{u}, x_1, \ldots, x_{l-1}, \bar{v})$ and $(\bar{v}, y_1, \ldots, y_{m-1}, \bar{w})$ with traversal times represented as $T_{\bar{u}\bar{v}}$ and $T_{\bar{v}\bar{w}}$ respectively, for all $z \in V$:*

$$V_s((\bar{u}, \ldots, \bar{v}, \ldots, \bar{w}))(z) = \frac{T_{\bar{u}\bar{v}} V_s((\bar{u}, \ldots, \bar{v}))(z) + T_{\bar{v}\bar{w}} V_s((\bar{v}, \ldots, \bar{w}))(z)}{T_{\bar{u}\bar{v}} + T_{\bar{v}\bar{w}}} \tag{17}$$

*Proof.* Let the travel time for the path $(\bar{u}, \ldots, \bar{v}, \ldots, \bar{w})$ be $T_{\bar{u}\bar{w}}$. It is easy to observe that $T_{\bar{u}\bar{w}} = T_{\bar{u}\bar{v}} + T_{\bar{v}\bar{w}}$ as we assume that the traversal time is independent of previous actions of the agent.

$$T_{\bar{v}\bar{w}} V_s((\bar{v}, \ldots, \bar{w}))(z) = \int_0^{T_{\bar{v}\bar{w}}} \mathcal{V}_z(t) dt = \int_{T_{\bar{u}\bar{v}}}^{T_{\bar{u}\bar{w}}} \mathcal{V}_z(t - T_{\bar{u}\bar{v}}) d(t - T_{\bar{u}\bar{v}} = \int_{T_{\bar{u}\bar{v}}}^{T_{\bar{u}\bar{w}}} \mathcal{V}_z(t) d(t) \tag{18}$$

Hence, we have that,

$$\begin{aligned}
T_{\bar{u}\bar{v}} V_s((\bar{u}, \ldots, \bar{v}))(z) + T_{\bar{v}\bar{w}} V_s((\bar{v}, \ldots, \bar{w}))(z) &= \int_0^{T_{\bar{u}\bar{v}}} \mathcal{V}_z(t) d(t) + \int_{T_{\bar{u}\bar{v}}}^{T_{\bar{u}\bar{w}}} \mathcal{V}_z(t) d(t) \\
&= \int_0^{T_{\bar{u}\bar{w}}} \mathcal{V}_z(t) d(t)
\end{aligned} \tag{19}$$

$\square$

The following is proof for Theorem 2

*Proof.* Using the fact that we have probability distribution on the edges $E_d$, we have,

$$\begin{aligned}
\sum_{(u,v) \in E_d} P((u, v)) &= \sum_{(u,v) \in E_d} \pi(u) P(u \to v) \\
&= \sum_{u \in V_d} \sum_{\{v | (u,v) \in E_d\}} \pi(u) P(u \to v) \\
&= \sum_{u \in V_d} \pi(u) \sum_{\{v | (u,v) \in E_d\}} \pi(u) P(u \to v) \\
&= \sum_{u \in V_d} \pi(u)
\end{aligned}$$

$\square$

**Theorem 5.** *Let us define the following for any given natural $n$, time $0 \leq t \leq T$, and $w \in V$:*

$$\mathcal{X}^i = \sum_{(u,v) \in E_d} h^i_{(u,v)}(t) V_s((u,v))(w)$$

*Then the following holds:*

$$\sum_{i=1}^n \mathcal{X}^i - \sum_{1 \leq i < j \leq n} \mathcal{X}^i \mathcal{X}^j \leq Y_t(w) \leq \sum_{i=1}^n \mathcal{X}^i$$

*Proof.* By definition of $h^i_{(u,v)}(t)$, only one of them can be 1 at a given time $t$ for any $1 \leq i \leq n$. That is, $\sum_{(u,v) \in E_d} h^i_{(u,v)}(t) = 1$

As $0 \leq V_s((u,v)) \leq 1$ for any $(u,v) \in E_d$, we have $0 \leq \mathcal{X}^i \leq 1$

For $n = 1$, it is easy to observe that the equality holds. We now prove the result using induction. Assume that the statement is true for $n$.

$$\implies \sum_{i=1}^n \mathcal{X}^i - \sum_{1 \leq i < j \leq n} \mathcal{X}^i \mathcal{X}^j \leq 1 - \prod_{i=1}^n \left(1 - \mathcal{X}^i\right) \leq \sum_{i=1}^n \mathcal{X}^i \tag{20}$$

When there are $n + 1$ agents, we have the following:

$$1 - \prod_{i=1}^{n+1} \left(1 - \mathcal{X}^i\right) = 1 - \left(1 - \mathcal{X}^{n+1}\right) \prod_{i=1}^n \left(1 - \mathcal{X}^i\right) \leq 1 - \left(1 - \mathcal{X}^{n+1}\right) \left(1 - \sum_{i=1}^n \mathcal{X}^i\right)$$

$$\implies 1 - \prod_{i=1}^{n+1} \left(1 - \mathcal{X}^i\right) \leq \sum_{i=1}^{n+1} \mathcal{X}^i - \sum_{i=1}^n \mathcal{X}^i \mathcal{X}^{n+1} \leq \sum_{i=1}^{n+1} \mathcal{X}^i$$

$$\tag{21}$$

Let $\mathcal{M}_1 = \sum_{i=1}^n \mathcal{X}^i$ and $\mathcal{M}_2 = \sum_{1 \leq i < j \leq n} \mathcal{X}^i \mathcal{X}^j$. Then, we have

$$1 - \prod_{i=1}^{n+1} \left(1 - \mathcal{X}^i\right) = 1 - \left(1 - \mathcal{X}^{n+1}\right) \prod_{i=1}^n \left(1 - \mathcal{X}^i\right) \geq 1 - \left(1 - \mathcal{X}^{n+1}\right) \left(1 - \mathcal{M}_1 + \mathcal{M}_2\right)$$

$$\implies 1 - \prod_{i=1}^{n+1} \left(1 - \mathcal{X}^i\right) \geq \mathcal{X}^{n+1} + \mathcal{M}_1 - \mathcal{M}_2 + \left(\mathcal{X}^{n+1}\right) \left(\mathcal{M}_2 - \mathcal{M}_1\right)$$

$$\geq \left(\mathcal{X}^{n+1} + \mathcal{M}_1\right) - \left(\mathcal{M}_2 + \mathcal{M}_1 \mathcal{X}^{n+1}\right) \geq \sum_{i=1}^{n+1} \mathcal{X}^i - \sum_{1 \leq i < j \leq n+1} \mathcal{X}^i \mathcal{X}^j$$

$$\tag{22}$$

$\square$

Now we prove Theorem 1 using the above result.

*Proof.* We first show the relation between $\mathcal{X}^i(t)$ and the probability values.

$$\mathcal{X}^i(t) = \sum_{(u,v) \in E_d} h^i_{(u,v)}(t) V_s((u,v))(w)$$

$$\implies \int_0^T \mathcal{X}^i(t) dt = \int_0^T \sum_{(u,v) \in E_d} V_s((u,v))(w) h^i_{(u,v)}(t) dt$$

$$= \sum_{(u,v) \in E_d} V_s((u,v))(w) \int_0^T h^i_{(u,v)}(t) dt \qquad (23)$$

$$\implies \sum_{i=1}^n \int_0^T \mathcal{X}^i(t) dt = nT \sum_{(u,v) \in E_d} P((u,v)) V_s((u,v))(w)$$

Using Theorem 5, we have the following:

$$\sum_{i=1}^n \mathcal{X}^i(t) - \sum_{1 \le i < j \le n} \mathcal{X}^i(t) \mathcal{X}^j(t) \le Y_t(w) \le \sum_{i=1}^n \mathcal{X}^i(t)$$

Hence, we have the following:

$$\int_0^T Y_t(w) \, dt \le nT \sum_{(u,v) \in E_d} P((u,v)) V_s((u,v))(w)$$

We also have the following:

$$nT \sum_{(u,v) \in E_d} P((u,v)) V_s((u,v))(w) \le \sum_{1 \le i < j \le n} \int_0^T \mathcal{X}^i(t) \mathcal{X}^j(t) \, dt + \int_0^T Y_t(w) \, dt$$

We can write $Y_t(w)$ in terms of $\mathcal{X}^i(t)$ as follows:

$$Y_t(w) = 1 - \prod_{i=1}^n \left( 1 - \mathcal{X}^i(t) \right) \qquad (24)$$

Using the inequality of geometric mean and arithmetic mean, we can write:

$$\left( 1 - \frac{\sum_{i=1}^n \mathcal{X}^i(t)}{n} \right)^n \ge \prod_{i=1}^n \left( 1 - \mathcal{X}^i(t) \right) \qquad (25)$$

As $0 \le \mathcal{X}^i(t) \le 1$, we have that $0 \le 1 - \left( \sum_{i=1}^n \mathcal{X}^i(t) \right) / n \le 1$.

$$\implies 1 - \frac{\sum_{i=1}^n \mathcal{X}^i(t)}{n} \ge \prod_{i=1}^n \left( 1 - \mathcal{X}^i(t) \right)$$

$$\implies \frac{\sum_{i=1}^n \mathcal{X}^i(t)}{n} \le Y_t(w) \qquad (26)$$

$$\implies T \sum_{(u,v) \in E_d} P((u,v)) V_s((u,v))(w) \le \int_0^T Y_t(w) dt$$

$\square$

We can define the probability that a particular node $w$ is visible as any given time $t$ chosen uniformly at random in the interval $[0, T]$ in terms of $Y_t(w)$ as follows:

$$P_V(w) = \frac{1}{T} \int_0^T Y_t(w) dt \qquad (27)$$

Let us make binary indicator variables for each node represented as $V_w$. Then the probability that $V_w = 1$ for any time time is $P_V(w)$. Liu et al. (2010) defines the common information using an

auxiliary random variable $W$ which makes individual variables independent. We can now write the common information as:

$$C(\{V_w|w \in V\}) = \inf I(\{V_w|w \in V\}; W) \tag{28}$$

We also have the following Wyner (1975):

$$I(\{V_w|w \in V\}; W) = H(\{V_w|w \in V\}) - H(\{V_w|w \in V\}|W)$$
$$\implies I(\{V_w|w \in V\}; W) = H(\{V_w|w \in V\}) - \sum_{w \in V} H(V_w|W) \tag{29}$$

where $H$ is the joint entropy of the random variables.

Notice that we can select $W$ such that $P((u, v))$ for all $(u, v) \in E_d$ become independent which also implies that $P_V(w)$ for all $w \in V$ become independent given $W$. Using Theorem 1, we have the following:

$$H(V_w|W) \leq H(V_w|\{P((u, v))|(u, v) \in E_d\})$$
$$\implies H(V_w|W) \leq -P_V(w) \log P_V(w)$$
$$\implies H(V_w|W) \leq -P_V(w) \log \mathcal{A}(w)$$
$$\implies H(V_w|W) \leq -n\mathcal{A}(w) \log \mathcal{A}(w) \tag{30}$$
$$\implies I(\{V_w|w \in V\}) \geq H(\{V_w|w \in V\}) - n \sum_{w \in V} -\mathcal{A}(w) \log \mathcal{A}(w)$$

where $\mathcal{A}(w) = \sum_{(u,v) \in E_d} P((u, v)) V_s((u, v))(w)$. This implies that minimizing $\sum_{w \in V} -\mathcal{A}(w) \log \mathcal{A}(w)$ will maximize the common information.

## B   FINE GRAINED ESTIMATOR

We now give the proof for Theorem 3 below.

*Proof.* Let us define a path $(\bar{u}, w_1, w_2, \ldots, w_{l-1}, \bar{v})$ such that $(w_i, w_{i+1}) \in E_d$ for all $0 \leq i \leq l-1$ where $w_0 = \bar{u}$ and $w_l = \bar{v}$. As a shorthand, we represent the path as $(\bar{u}, \ldots, \bar{v})$.

Let us define a modified edge set $\bar{E}_d = E_d \cup \{(\bar{u}, \ldots, \bar{v})\}$ and also have a probability distribution $\bar{P}((u, v))$ over $\bar{E}_d$ analogous to the distribution $P((u, v))$ over $E_d$. The only change is that $\bar{P}((\bar{u}, \ldots, \bar{v}))$ gives the probability that the path is being traversed. We can now define the functions $\bar{h}_e^i(t)$ for all $e \in \bar{E}_d$. We now define:

$$\bar{P}(e) = \frac{1}{nT} \sum_{i=1}^{n} \int_0^T \bar{h}_e^i(t) dt \quad \forall e \in \bar{E}_d \tag{31}$$

and, we will now show that it is indeed a proper distribution.

Define indicator functions $g_e^i(t)$ for all $e \in E_d$ and $1 \leq i \leq n$ such that $g_e^i(t) = 1$ when agent $i$ is traversing the edge $e$ while the agent is traversing the path $(\bar{u}, \ldots, \bar{v})$. Obviously, for any edge $e$ that is not in the path, $g_e^i(t) = 0$ for all $0 \leq t \leq T, 1 \leq i \leq n$. Furthermore,

$$h_e^i(t) = \bar{h}_e^i(t) + g_e^i(t) \quad \forall e \in E_d \tag{32}$$

$$\implies P(e) = \frac{1}{nT} \sum_{i=1}^{n} \int_0^T \bar{h}_e^i(t) + g_e^i(t) \, dt \quad \forall e \in E_d$$

$$\implies P(e) = \bar{P}(e) + \frac{1}{nT} \sum_{i=1}^{n} \int_0^T g_e^i(t) \, dt \quad \forall e \in E_d$$

$$\implies 1 = \sum_{e \in E_d} \bar{P}(e) + \frac{1}{nT} \sum_{i=1}^{n} \int_0^T \sum_{e \in E_d} g_e^i(t) \, dt$$

$$\implies 1 = \sum_{e \in E_d} \bar{P}(e) + \frac{1}{nT} \sum_{i=1}^{n} \int_0^T \bar{h}_{(\bar{u},\dots,\bar{v})}^i(t) \, dt$$

$$\implies 1 = \sum_{e \in \bar{E}_d} \bar{P}(e)$$

Suppose that we define $T_e$ as the time spent on the edge $e \in E_d$ while traversing the path once. It is easy to observe that $\sum_{e \in E_d} T_e = T_{\bar{u}\bar{v}}$. We can use the time to get the following relation:

$$\int_0^T \sum_{i=1}^{n} g_e^i(t) \, dt = \frac{T_e}{T_{\bar{u}\bar{v}}} \int_0^T \sum_{i=1}^{n} \bar{h}_{(\bar{u},\dots,\bar{v})}^i(t) \, dt \; \forall e \in E_d \tag{33}$$

$$\implies P(e) = \bar{P}(e) + \frac{T_e}{T_{\bar{u}\bar{v}}} \bar{P}((\bar{u},\dots,\bar{v})) \; \forall e \in E_d$$

Now we show that the loss function over the probabilities on $\bar{E}_d$ is the same for the probabilities on $E_d$. It is easy to observe that the loss will be the same if we can show that $\sum_{e \in E_d} V_s(e)P(e) = \sum_{e \in \bar{E}_d} V_s(e)\bar{P}(e)$. Then, by using Lemma 1, we have,

$$\sum_{e \in \bar{E}_d} V_s(e)\bar{P}(e) = \sum_{e \in E_d} V_s(e)\bar{P}(e) + V_s((\bar{u},\dots,\bar{v}))\bar{P}((\bar{u},\dots,\bar{v}))$$

$$= \sum_{e \in E_d} V_s(e)\bar{P}(e) + \bar{P}((\bar{u},\dots,\bar{v})) \sum_{e \in E_d} \frac{T_e}{T_{\bar{u}\bar{v}}} V_s(e)$$

$$= \sum_{e \in E_d} V_s(e) \left( \bar{P}(e) + \frac{T_e}{T_{\bar{u}\bar{v}}} \bar{P}((\bar{u},\dots,\bar{v})) \right)$$

$$= \sum_{e \in E_d} V_s(e)P(e)$$

$\square$

The following is a proof for Theorem 4.

*Proof.* First we need to show that $\bar{P}$ and $\hat{P}$ form a probability distribution over the edges. Let $q = 1 - p$. Notice that $\bar{h}_{(u,v)}^i(t) = I(t)h_{(u,v)}^i(t)$ for all $0 \le t \le T$ and $(u,v) \in E_d$.

$$\implies \bar{P}((u,v)) = \frac{1}{npT} \int_0^T I(t)h_{(u,v)}^i(t)dt$$

$$\implies \sum_{(u,v) \in E_d} \bar{P}((u,v)) = \frac{1}{npT} \sum_{i=1}^{n} \int_0^T I(t)dt \tag{34}$$

$$\implies \sum_{(u,v) \in E_d} \bar{P}((u,v)) = 1$$

From the way $\bar{P}$ and $\hat{P}$ is defined, it is easy to observe that,

$$P((u,v)) = p\bar{P}((u,v)) + q\hat{P}((u,v)) \quad \forall (u,v) \in E_d$$
$$\implies 1 = p + q \sum_{(u,v)\in E_d} \hat{P}((u,v)) \tag{35}$$
$$\implies \hat{P}((u,v)) = 1$$

Hence, we have that both are probability distributions over the edges. Therefore, we can way that it is possible to decompose $\bar{P}((u,v))$ into $\bar{\pi}(u)$ and $\bar{P}(u \to v)$, and $\hat{P}((u,v))$ into $\hat{\pi}(u)$ and $\hat{P}(u \to v)$.

For any $u \in V_d$, we have the following:

$$\implies \pi(u) = \sum_{\{v|(u,v)\in E_d\}} p\bar{P}((u,v)) + q\hat{P}((u,v)) \tag{36}$$
$$\implies \pi(u) = p\bar{\pi}(u) + q\hat{\pi}(u)$$

Let us define the matrix $M$ of size $|V_d| \times |V_d|$ and a vector $x$ of size $|V_d|$ which is indexed by the vertices in $V_d$ such that:

$$M(v,u) = \begin{cases} P(u \to v) & (u,v) \in E_d \\ 0 & otherwise \end{cases} \tag{37}$$
$$x(u) = \pi(u)$$

Similarly, let us also define the following:

$$\bar{M}(v,u) = \begin{cases} \bar{P}(u \to v) & (u,v) \in E_d \\ 0 & otherwise \end{cases} \qquad \bar{x}(u) = \bar{\pi}(u)$$
$$\hat{M}(v,u) = \begin{cases} \hat{P}(u \to v) & (u,v) \in E_d \\ 0 & otherwise \end{cases} \qquad \hat{x}(u) = \hat{\pi}(u) \tag{38}$$

Using equation 36, we have that $x = p\bar{x} + q\hat{x}$, and equation 35 implies that $Mx = p\bar{M}\bar{x} + q\hat{M}\hat{x}$.

$$\implies x - Mx = p(\bar{x} - \bar{M}\bar{x}) + q(\hat{x} - \hat{M}\hat{x})$$
$$\implies (I - M)x = p(I - \bar{M})\bar{x} + q(I - \hat{M})\hat{x} \tag{39}$$

Hence, it is sufficient to constraint that $\bar{M}\bar{x} = \bar{x}$ and $\hat{M}\hat{x} = \hat{x}$ which is same as saying:

$$\sum_{\{v|(u,v)\in E_d\}} \hat{\pi}(u)\hat{P}(u \to v) = \hat{\pi}(v) \quad \text{and} \quad \sum_{\{v|(u,v)\in E_d\}} \bar{\pi}(u)\bar{P}(u \to v) = \bar{\pi}(v)$$

$\square$

## C  LARGE ENVIRONMENT EXPERIMENTS

The large environment is a $70 \times 70$ grid-like structure with an uneven but smooth topology shown in Figure 5. Two stark differences from the small environment are, 1) there are no walls that suddenly clip the visibility, and 2) the topology does not restrict the movement of agents. We do simulations for homogeneous teams with 30 agents with unidirectional visibility.

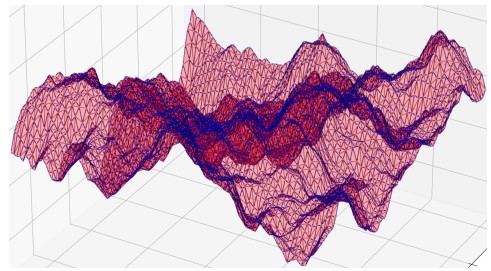

Figure 5: The large environment has a variety of features that are observed in real world topographies. It has multiple different hills and valleys resulting in a complex situation to analyze even for human experts.

Figure 7 shows the cumulative counts and Figure 6 shows the total variation distance of the distribution approximated by the trajectories and the distribution found using BOIL process.

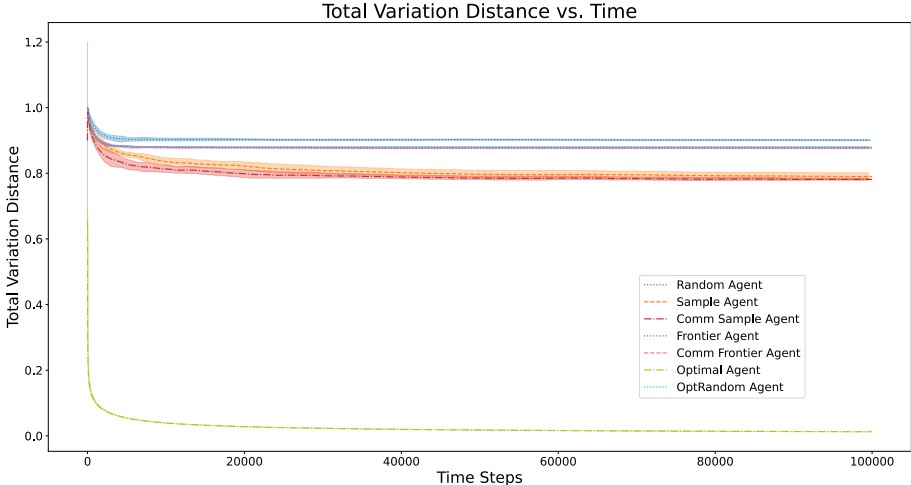

Figure 6: The figure depicts Time steps vs. Total variation distance between the approximate distribution of the agent trajectories and the distribution found using the BOIL process for the large environment. **BOIL Distribution Strategies:** *Sample Agent, Comm Sample Agent, Optimal Agent*

Let us initially examine Figure 6 to assess the convergence rate. It becomes evident that random walks and frontier-based exploration strategies converge to a similar distance value. Conversely, Sample and Comm Sample strategies struggle to achieve convergence. This disparity is further underscored by the visibility counts depicted in Figure 7. The challenge of attaining convergence to the desired distribution becomes apparent as the environment's scale and complexity increase. However, the visibility counts also suggest that coverage remains satisfactory once agents reach the distribution obtained through the BOIL process. Thus, despite the inherent difficulty in achieving convergence, the obtained distribution is effective for ensuring adequate coverage.

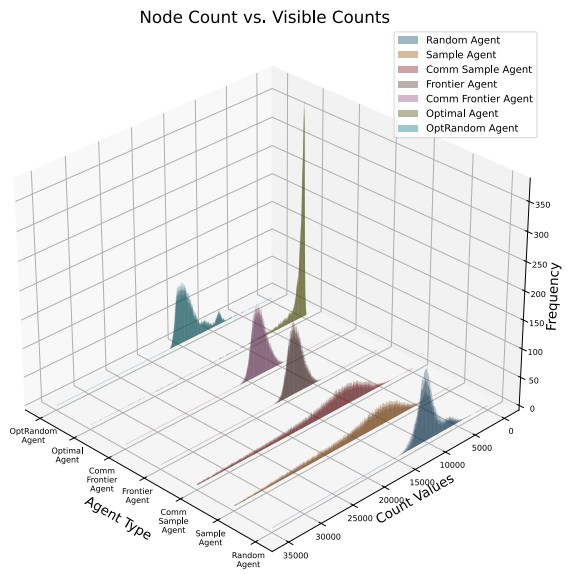

Figure 7: Similar to Figure 2, this figure shows the same type of data for the large environment. Z-axis illustrates the counts of visible nodes corresponding to values on the Y-axis, while the X-axis denotes the types of homogeneous agents considered in the experiment. Curves' opacity signifies variations in values, with lower opacity indicating the upper end of the variance and higher opacity representing the lower end for frequencies. The central divider indicates the mean frequencies. **BOIL Distribution Strategies:** *Sample Agent, Comm Sample Agent, Optimal Agent*

It should be noted that despite the apparent plateauing of convergence, closer inspection of the raw values reveals a downward gradient for both the small and large environments for Sample and Comm Sample strategies. This observation aligns with theoretical expectations, as the sampling process theoretically guarantees convergence to the desired distribution. Furthermore, bidirectional paths in

the environment may contribute to oscillatory behavior, given the simplicity of the path-planning process.

A closer look at Figure 7 suggests that the distributions of Sample and Comm Sample agents exhibit a skewed peak resembling that of Optimal agents. This indicates that despite encountering challenges in achieving full convergence, these strategies strive to emulate the optimal distribution pattern.

