# OpenReview forum: "BOIL: Learning Environment Personalized Information"
_ICLR.cc/2025/Conference — Submitted to ICLR 2025_

### Official Review · Reviewer_Psai · 2024-11-01

**Soundness:** 3
**Presentation:** 3
**Contribution:** 3
**Rating:** 6
**Confidence:** 3

**Summary:**

The paper introduces BOIL (Blackbox Oracle Information Learning), a process designed to enhance multi-agent systems by extracting information from a blackbox oracle to improve agent behavior in complex environments. It leverages the Pagerank algorithm and common information maximization techniques to guide agents in tasks such as coverage, patrolling, and reachability. The authors demonstrate that BOIL outperforms heuristic-based strategies in extended-time simulations and complex environments, providing a scalable approach that remains independent of the number of agents.

**Strengths:**

1. BOIL (Blackbox Oracle Information Learning) offers an innovative method for addressing challenges in multi-agent systems by utilizing a blackbox oracle to extract valuable information about the environment. The originality lies in its novel use of established algorithms like Pagerank to solve complex tasks such as coverage, patrolling, and stochastic reachability in multi-agent systems. This is a significant departure from existing methods and introduces new possibilities for long-term strategy generation in dynamically changing environments.
2. The paper presents rigorous theoretical formulations to justify its claims. The mathematical grounding of BOIL in Pagerank and non-reversible Markov chains demonstrates a deep understanding of the problem. By framing the problem of agent task execution as a flow constraint problem and using common information maximization, the authors present a technically sophisticated solution that ensures the system remains scalable and efficient, regardless of the number of agents involved.
3. One of the standout strengths of BOIL is its scalability. Many existing multi-agent systems face challenges when trying to scale with the number of agents or the complexity of the environment. BOIL circumvents this issue by extracting and processing information in a way that is independent of the number of agents, allowing it to perform efficiently even in environments with a large number of agents.
4. The simulations show that BOIL can generate strategies that lead to better coverage and visibility in challenging environments, making it applicable to real-world use cases in fields such as autonomous robotics, distributed sensor networks, and security systems. Moreover, the fact that BOIL is able to function without requiring a large number of agents adds to its practical appeal, especially in resource-constrained systems.

**Weaknesses:**

1. The experiments primarily focus on the coverage problem, leaving patrolling and reachability tasks underexplored. The lack of direct validation for patrolling and reachability leaves open questions about how well BOIL can adapt to these different problem spaces, particularly in dynamic or more constrained environments. Including these experiments would better illustrate the full breadth of BOIL’s potential applications.
2. The convergence issues of the Sample and Comm Sample agents are discussed, but a deeper analysis is needed to understand why they fail to achieve optimal performance compared to the Optimal agent. For example, it would be valuable to understand whether these convergence issues stem from limitations in the algorithm's design, insufficient exploration of the environment, or perhaps an inherent trade-off in computational efficiency versus accuracy.
3. Although the paper claims that BOIL scales independently of the number of agents, it does not provide explicit scalability metrics or computational complexity analysis, particularly as the size of the environment or number of agents increases. The paper would benefit from a more detailed discussion of the computational complexity of BOIL, especially in larger environments, and explicit time complexity metrics would enhance its practical impact.
4. Certain sections of the paper, especially the Fine-Grained Estimation section, are mathematically dense and could benefit from clearer explanations. The current presentation, while technically correct, may be challenging for readers not intimately familiar with the underlying mathematical concepts. Including more intuitive examples or diagrams would make this section more accessible. This would not only improve the readability of the paper but also help a broader audience appreciate the nuances of the proposed techniques.

**Questions:**

1. Can the authors provide more experimental evidence for patrolling and reachability tasks to better demonstrate BOIL's generalizability?
2. Could you elaborate on why the Sample and Comm Sample agents struggle to converge in large environments? Are there any potential improvements or optimizations that could address this?
3. Are there specific scalability limits for BOIL when applied to very large agent numbers or more complex environments? Including some real-world use cases or benchmarks would strengthen this point.

---

> ### Author Response · Authors · 2024-11-16
>
> We thank the reviewer for their thoughtful comments.
>
> After considering the reviews from all reviewers, we have made the following broad changes:
> - The exact contributions of the work were not clear. We have made the contributions explicit in the introduction. The related works section has been largely rewritten to highlight the gap in existing literature and where our contributions fit to fill in the gap
> - It was hard to link the abstractions with real world scenarios. To address this concern, we have made various changes across the entire paper detailing the choice of abstraction and how it helps model real world conditions.
> - It was hard to link the theorems with the text. To resolve this, we have added context to the theorem of where it fits in the entire framework more explicitly.
>
> For your ease, the changes are highlighted in blue.
>
> Before addressing the comments, we want to let the reviewer know that Section 6 is converted into a discussion section with conclusions and future work as a subsection. Subsection 4.3 (Extensions to Patrolling and Reachability) has been shifted to Section 6 as it might confuse the reader what are the contributions, and as such the entire Section is highlighted even though the major change is only reordering the content of the original submitted copy.
>
> ## Addressing the Weaknesses
>
> > The experiments primarily focus on the coverage problem, leaving patrolling and reachability tasks underexplored. The lack of direct validation for patrolling and reachability leaves open questions about how well BOIL can adapt to these different problem spaces, particularly in dynamic or more constrained environments. Including these experiments would better illustrate the full breadth of BOIL’s potential applications.
>
> We agree with the reviewer that having the experiments would provide direct validation. However, because of the following challenges, it is hard to include these experiments:
> - Patrolling and reachability both are highly dependent on the control algorithm used as compared to coverage. As mentioned in the paper, patrolling is highly dependent on control as the number of points to be patrolled can be very low. So even if the agent goes closer to the point but is not able to cover the exact patrolling point, then it will not be considered patrolled. Such situations can easily happen when the control policy of the agent is not good. A similar case can be made for reachability as the agent needs to deviate from its path to actually reach the point where say an intruder is detected and may turn into a pursuit-evasion game. Furthermore, it also requires a different game setup to simulate such scenarios.
> - Both the problems themselves have a plethora of literature. It will be hard to do a fair evaluation for them without specifically focusing on the particular problem.
> - The main reason for introducing the problems was to showcase the flexibility of BOIL. As mentioned in related works, previous works themselves draw relations between the problems. To ensure that the reader understands our contributions experimentally are limited to coverage, we have shifted the extensions to the discussion section.
>
> > The convergence issues of the Sample and Comm Sample agents are discussed, but a deeper analysis is needed to understand why they fail to achieve optimal performance compared to the Optimal agent. For example, it would be valuable to understand whether these convergence issues stem from limitations in the algorithm's design, insufficient exploration of the environment, or perhaps an inherent trade-off in computational efficiency versus accuracy.
>
> The issues with convergence we believe come primarily from the limitations in control and the limited amount of information extracted from the oracle. To extract more information, we would require more computation and use Fine grained estimation introduced in Subsection 4.3
>
> The changes made in Subsection 4.2 will help the reader to understand the above:
> “Note how we have only constrained the flow of probability as Pagerank only ensures the global balanced condition. The oracle is supposed to give continuous paths but we solved for only the softer probabilistic constraint. In essence, running Algorithm 1 allows us to access some information from the oracle. While previous work does not require an oracle, they use hard constraints, thereby restricting custom control over the design trade-offs. Using the oracle formulation allows us to do fine-grained estimation.”
>
> We agree with the reviewer that further analysis is warranted but we believe that more mathematical tools need to be developed to do this analysis.

---

> ### Author Response · Authors · 2024-11-16
>
> > Although the paper claims that BOIL scales independently of the number of agents, it does not provide explicit scalability metrics or computational complexity analysis, particularly as the size of the environment or number of agents increases. The paper would benefit from a more detailed discussion of the computational complexity of BOIL, especially in larger environments, and explicit time complexity metrics would enhance its practical impact.
>
> The BOIL process extracts information as distributions which are truly independent of the agent count. However, the amount of information extracted is the same. How to ensure that it is properly used by multiple agents is something that depends on outside factors like communication limitations, onboard computation of the agents as well as the control policy. The reason why we highlight its independence on agent count is because previous works required defining agent count beforehand. In contrast, having information about the group strategy is more helpful if suddenly the agent counts increased or decreased. In the setting that we consider, the found distribution is something that is given to each agent. Individual agents then follow a simple uniform sampling process. The complexity now depends on what sampling algorithm is used. It should be noted that the complexity needs to be considered in a distributed computation model as each agent is supposed to have its own compute.
>
> Handling complexity with environment size is a trickier problem, primarily because of the following reasons:
> - The algorithm is an iterative process that uses approximate results. Furthermore, pagerank itself is also an iterative algorithm. The number of iterations it will require to stop to get within a certain threshold is not a trivial problem to solve as it is a function of the structure of the graph and the initial estimate. Bogolubsky et. al. (2016), whose algorithm we use, give the complexity in terms of number of arithmetic operations required as a function of step size, error in gradient calculation, number of iterations and the final error.
> - The algorithm is inherently made for distributed computation and it makes more sense to talk about the complexity in a distributed computing paradigm.
> - Because of the nested nature, it is possible that the pagerank converges quickly as the perturbations made in the main iterative loop are small.
> - The complexity is also a function of how the graph is stored, computation of visibility and the computation of loss function.
>
> > Certain sections of the paper, especially the Fine-Grained Estimation section, are mathematically dense and could benefit from clearer explanations. The current presentation, while technically correct, may be challenging for readers not intimately familiar with the underlying mathematical concepts. Including more intuitive examples or diagrams would make this section more accessible. This would not only improve the readability of the paper but also help a broader audience appreciate the nuances of the proposed techniques.
>
> We have added some more material in the revised version of the paper that draws connections between the two theorems and the base formulation presented for coverage.
>
> ## Addressing the questions:
>
> > Can the authors provide more experimental evidence for patrolling and reachability tasks to better demonstrate BOIL's generalizability?
>
> As mentioned previously, a fair evaluation would be hard.
>
> > Could you elaborate on why the Sample and Comm Sample agents struggle to converge in large environments? Are there any potential improvements or optimizations that could address this?
>
> As mentioned in the paper, the simple sampling strategy is not a good control policy. It is possible that because of hard continuity constraints, which are not a part of the original sampling process, the agents keep moving in small neighborhoods and slowly make their way towards the entire available area to cover.
>
> In such a large environment, coordination between agents makes a huge difference as the number of agents is far less as compared to the size of the environment. We use a simple communication strategy which only aggregates the covering from individual agents to slightly bias the sampling process.
>
> Better communication strategies and control policy can go a long way to address this issue.
>
> > Are there specific scalability limits for BOIL when applied to very large agent numbers or more complex environments? Including some real-world use cases or benchmarks would strengthen this point.
>
> As mentioned before, BOIL process itself is independent of agent count. Complexity of the environment can affect the time required to compute the pagerank as well as the overall number of iterations required to find a good solution.

---

> > ### Comment · Reviewer_Psai · 2024-11-25
> > **Thank the authors for the feedback.**
> >
> > After reading the feedback, some concerns like the BOIL scales have been addressed. I have also read other reviewers’ comments, and I would like to keep my score.

---

### Official Review · Reviewer_w3GN · 2024-11-03

**Soundness:** 3
**Presentation:** 2
**Contribution:** 2
**Rating:** 5
**Confidence:** 2

**Summary:**

The paper considers the problem of one or more agents learning from an environment as they traverse it. It uses an algorithm based on PageRank to extract information from this traversal of the environment.

**Strengths:**

The problem defined is very general, and paper pulls together ideas from several lines of work spread over multiple periods of time in order to arrive at its underlying approach. This unification of disparate approaches and lines of work is a contribution of the work, alongside the algorithms themselves.

**Weaknesses:**

The paper's level of general abstraction is helpful in being able to talk about the connection to disparate earlier lines of work, but it comes with the counterbalancing problem that it makes it hard to identify where the concrete improvements come from in the current approach.

In particular:

- Can the paper specify one or more canonical application domains where the approach would be most likely to be applied? As it stands, even the computational experiments take place in abstract environments that make it hard to see their mapping onto real applications.

- How do the agent strategies in Section 5.1 relate to the earlier approaches from related work? Other than the reference to Yamauchi (1997, 1998), there are no other links between the baseline strategies in this subsection and the description of what's come before in earlier work.

- More generally, what are the concrete ways in which the approaches developed in this paper improve on earlier approaches?

**Questions:**

It would be helpful if the authors could address the weaknesses listed above.

---

> ### Author Response · Authors · 2024-11-16
>
> We thank the reviewer for their thoughtful comments.
>
> After considering the reviews from all reviewers, we have made the following broad changes:
> - The exact contributions of the work were not clear. We have made the contributions explicit in the introduction. The related works section has been largely rewritten to highlight the gap in existing literature and where our contributions fit to fill in the gap
> - It was hard to link the abstractions with real world scenarios. To address this concern, we have made various changes across the entire paper detailing the choice of abstraction and how it helps model real world conditions.
> - It was hard to link the theorems with the text. To resolve this, we have added context to the theorem of where it fits in the entire framework more explicitly.
>
> For your ease, the changes are highlighted in blue.
>
> Before addressing the comments, we want to let the reviewer know that Section 6 is converted into a discussion section with conclusions and future work as a subsection. Subsection 4.3 (Extensions to Patrolling and Reachability) has been shifted to Section 6 as it might confuse the reader what are the contributions, and as such the entire Section is highlighted even though the major change is only reordering the content of the original submitted copy.
>
> > Can the paper specify one or more canonical application domains where the approach would be most likely to be applied? As it stands, even the computational experiments take place in abstract environments that make it hard to see their mapping onto real applications.
>
> We have added a line in the introduction to explicitly clarify that many real-world problems like mobile sensor coverage, forest fire detection, agricultural monitoring and traffic data collection can be modeled using coverage. In addition, to further ensure that the link with the abstraction choices and real-world difficulties is clearly established, we have added more material in Subsection 4.1. To summarize the changes, the directed graph structure allows us to model one-ways and fire exits in urban settings, and model restrictions that come due to the limitations on the power provided by motors for robot movement. Furthermore, it also allows us to model the visibility of ground vehicles that cannot see in all directions.
>
> > How do the agent strategies in Section 5.1 relate to the earlier approaches from related work? Other than the reference to Yamauchi (1997, 1998), there are no other links between the baseline strategies in this subsection and the description of what's come before in earlier work.
>
> With the modifications primarily in the related works section, it becomes clear that we are addressing some gaps in modeling itself which makes it difficult to do a fair evaluation of BOIL with other methods. We explicitly mention the issue in the “Experiments & Results” section as follows:
> “Our experiments model agents with unidirectional visibility and irreversible movement which is a limitation for many previous works even though both conditions are found ubiquitously in real world situations. Works that consider general agent movement and visibility become intractable for the size of environment and agent count that we consider. To our knowledge, no baseline provides a fair comparison for evaluation. Hence, we select a few strategies that fit the requirements of the task.”
>
> > More generally, what are the concrete ways in which the approaches developed in this paper improve on earlier approaches?
>
> - There is a fundamental tradeoff between solution optimality, computational tractability, and scalability in environment size and agent counts for coverage. Earlier approaches try to balance all of them but do not provide a fine-grained control on these trade-offs. We have made changes in the introduction so that it is explicitly mentioned that this work aims to address this issue.
> - We introduce the concept of oracles and optimization based on softer probabilistic constraints to facilitate the fine-grained control for various tradeoffs. We have added more explanation near the end of Subsection 4.2 (Solving as Flow Constraint Problem) to ensure clarity for the reader:
> “Note how we have only constrained the flow of probability as Pagerank only ensures the global balanced condition. The oracle is supposed to give continuous paths but we solved for only the softer probabilistic constraint. In essence, running Algorithm 1 allows us to access some information from the oracle. While previous work does not require an oracle, they use hard constraints, thereby restricting custom control over the design trade-offs. Using the oracle formulation allows us to do fine-grained estimation.”
> - We discuss how it is possible to formulate other problems like patrolling and stochastic reachability using the same framework. Previous works had drawn the connection but did not discuss how it is possible.

---

> > ### Comment · Reviewer_w3GN · 2024-11-24
> >
> > I appreciate the replies by the authors, which are helpful in particular for understanding more about the context of the work. I still think the paper would benefit by making the connection to motivating applications more clear; while it is useful to mention mobile sensor coverage, forest fire detection, agricultural monitoring and traffic data collection at the outset, it would still help in justifying the high level of abstraction in the current formulation to see a comparison to specific approaches in one or more of these domains. For example, does the high level of abstraction here help in identifying new approaches that weren't apparent in more domain-specific approaches, and do the resulting methods offer improvements? All of these would be valuable additions to the evaluation of the method.  Given this, I prefer to leave my score where it is.

---

### Official Review · Reviewer_FTNz · 2024-11-04

**Soundness:** 3
**Presentation:** 3
**Contribution:** 2
**Rating:** 5
**Confidence:** 2

**Summary:**

This paper proposes an Algorithm named BOIL (Blackbox Oracle Information Learning) for navigating complex environments in multi-agent systems. The BOIL assumes access to an oracle whose information is indirectly accessible and its objective is to devise a computationally scalable approach to extract insights from this oracle. Leveraging the Pagerank algorithm and common information maximization, BOIL facilitates the extraction of information to guide long-term agent behavior applicable to problems such as coverage, patrolling, and stochastic reachability. Extensive experiments validate the empirical performance of BOIL.

**Strengths:**

This paper is well written with a clear logical flow.

The proposed algorithm BOIL is complemented with rigorous theoretical analysis and extensive empirical experiments. Both theoretical insights and empirical insights are drawn.

The setting of having access to an oracle whose information is indirectly accessible looks novel.

**Weaknesses:**

The assumption of having access to an oracle whose information is indirectly accessible needs justification. The feasibility of this assumption in real-world assumption is unclear.

The contribution of this paper is not clear to me. In particular, how this work improves the SOTA is unclear. This paper considers the setting of having access to an oracle whose information is indirectly accessible. Should the main contribution be claimed as a novel setting? Hasn’t this setting been considered in previous literature? From a methodological point of view, the techniques used in this paper look conventional. Compared to SOTA techniques, it is unclear to me whether this makes paper makes any methodology contribution.

The related work should not just list key papers in literature. The difference between this work and previous works is not discussed. How this paper advances previous is also not discussed.

**Questions:**

See the weakness part.

---

> ### Author Response · Authors · 2024-11-16
>
> We thank the reviewer for their thoughtful comments.
>
> After considering the reviews from all reviewers, we have made the following broad changes:
> - The exact contributions of the work were not clear. We have made the contributions explicit in the introduction. The related works section has been largely rewritten to highlight the gap in existing literature and where our contributions fit to fill in the gap
> - It was hard to link the abstractions with real world scenarios. To address this concern, we have made various changes across the entire paper detailing the choice of abstraction and how it helps model real world conditions.
> - It was hard to link the theorems with the text. To resolve this, we have added context to the theorem of where it fits in the entire framework more explicitly.
>
> For your ease, the changes are highlighted in blue.
>
> Before addressing the comments, we want to let the reviewer know that Section 6 is converted into a discussion section with conclusions and future work as a subsection. Subsection 4.3 (Extensions to Patrolling and Reachability) has been shifted to Section 6 as it might confuse the reader what are the contributions, and as such the entire Section is highlighted even though the major change is only reordering the content of the original submitted copy.
>
> > The assumption of having access to an oracle whose information is indirectly accessible needs justification. The feasibility of this assumption in real-world assumption is unclear.
>
> The oracle creates continuous paths for the agents, which are supposed to be optimal. From the construction we follow in the paper, we observe that it is possible to create a probability distribution over the edges of the directed graph representing the environment. The algorithm we present allows us to estimate the probability distribution without directly having the paths. The important thing to note here is that the obtained probability distribution only contains a part of the information required to get the original paths that the oracle would have produced.
>
> Using this roundabout method of defining the overall process allows us to think about the fine grained estimation (Explained in subsection 4.3 in the modified version) as a process to extract more information about agent paths from the oracle. We have added more explanation near the end of Subsection 4.2 (Solving as Flow Constraint Problem) to ensure clarity for the reader:
> “Note how we have only constrained the probability flow as Pagerank only ensures the global balanced condition. The oracle is supposed to give continuous paths, but we solved it only for the softer probabilistic constraint. In essence, running Algorithm 1 allows us to access some information from the oracle. While previous work does not require an oracle, they use hard constraints, thereby restricting custom control over the design trade-offs. Using the oracle formulation allows us to do \textit{fine-grained estimation}.”
>
> > The contribution of this paper is not clear to me. In particular, how this work improves the SOTA is unclear. This paper considers the setting of having access to an oracle whose information is indirectly accessible. Should the main contribution be claimed as a novel setting? Hasn’t this setting been considered in previous literature? From a methodological point of view, the techniques used in this paper look conventional. Compared to SOTA techniques, it is unclear to me whether this makes paper makes any methodology contribution.
>
> - Given the complexity of the coverage task, there is a fundamental tradeoff between solution optimality, computational tractability, and scalability in environment size and agent counts. Earlier approaches try to balance all of them but do not provide a fine-grained control on these trade-offs. For example, assuming independence of agent strategies creates a tradeoff between scalability and optimality of solutions. We have made changes in the introduction so that it is explicitly mentioned that this work aims to address this issue.
> - We introduce the concept of oracles and optimization based on softer probabilistic constraints to facilitate the fine-grained control for various tradeoffs.
> - We discuss how it is possible to formulate other problems like patrolling and stochastic reachability using the same framework. Previous works had drawn the connection but did not discuss how it is possible. The changes in the Introduction as well as Related Works section addresses this and makes it clear to the reader that we will discuss how it is possible in our framework.
>
> > The related work should not just list key papers in literature. The difference between this work and previous works is not discussed. How this paper advances previous is also not discussed.
>
> As mentioned previously, we have largely rewritten the section to highlight the gap in the literature and where our work fits.

---

### Official Review · Reviewer_QVfK · 2024-11-06

**Soundness:** 2
**Presentation:** 1
**Contribution:** 2
**Rating:** 5
**Confidence:** 3

**Summary:**

The authors introduce the Blackbox Oracle Information Learning (BOIL) process, a scalable method for extracting insights about environment structure in multi-agent systems. BOIL uses the Pagerank algorithm and information theory to capture information on agents' long-term behavior, showing versatility across various tasks, such as coverage, patrolling, and stochastic reachability, by framing them as common information maximization problems. The adaptability of BOIL, particularly its potential for online updates, positions it as a promising tool for enhancing performance in complex environments. Despite challenges in control dependencies, the authors believe the proposed solution offers a valuable step toward more adaptable and high-performing multi-agent systems.

**Strengths:**

The proposed solution BOIL provides a scalable solution for multi-agent systems that can adapt to a variety of tasks, including coverage, patrolling, and stochastic reachability. BOIL’s use of the Pagerank algorithm and information theory for extracting long-term behavioral insights seems to offer a novel application of these techniques in multi-agent environments.

**Weaknesses:**

The presentation of the paper is poor, I struggled a lot but still had trouble understanding most of the technical part. For example,

1. The entire section 3, though clearly written by itself, is confusing to me as it is introduced before the problem formulation. I do not understand how the introduced technical tools like the non-reversible Markov chain and supervised pagerank relate to the problem we target to solve.
2. Section 4, where the problem formulation is supposed to be presented, is also confusing to me. If I understand it correctly, the problem statement is hidden in L.224, which states that "The coverage problem can be defined as minimizing the expected information from all the $Y_t$ for all $0\leq t\leq T$". This is not a rigorous mathematical statement and I still do not get what is the concrete problem we aim to solve from this sentence. I also do not understand what the purposes of Lemma 1 and Theorem 1 are. I guess the properties revealed by Lemma 1 and Theorem 1 are crucial in terms of understanding the problem formulation, but I failed to see the connection here.
3. Section 4.3 is supposed to introduce the definition of Patrilling and Reachability problem. I can see that compared to coverage problem, the patrolling problem is to simply replace the set $V$ to $V_p$ in the definition of $\mathcal{L}$. However, I do not understand the description about the reachability problem. It would be much clearer if putting the definitions of the three problems together at the beginning.
4. All the lemmas and theorems are introduced abruptly. Though full proof details are provided in appendix, no proof sketch or high-level overview is provided in the main paper. This writing approach makes readers extremely difficult to grasp the significance and implications of each proof, as well as the logical connections among the theorems and lemmas. Consequently, understanding the broader flow and purpose of the mathematical arguments are challenging.

Due to these issues, I do not think I can full get the main contribution of this work. I might have missed something and I hope the authors could clarify these and crystalize their main contribution in their response, in particular, what the problem is and what the guarantee of the proposed method BOIL has.

**Questions:**

Please address the concerns I raised in weaknesses.

1. In section 5, it is mentioned that "We do experiments for the coverage problem and not for the rest of the extensions mentioned in
subsection 4.3." Do you have any explanation or justification for this decision? Do the results on the coverage problem suggest any meaningful observation for the other two extensions?
2. What is the rationale behind the 7 choices of the agent types in section 5.1? Are they commonly used in the literature?
3. The experiments are based on a synthetic environment. I'm wondering if this is a common benchmark and whether there are more realistic problem sets to test the performance of BOIL.

---

> ### Author Response · Authors · 2024-11-16
>
> We thank the reviewer for their thoughtful comments.
>
> After considering the reviews from all reviewers, we have made the following broad changes:
> - The exact contributions of the work were not clear. We have made the contributions explicit in the introduction. The related works section has been largely rewritten to highlight the gap in existing literature and where our contributions fit to fill in the gap
> - It was hard to link the abstractions with real world scenarios. To address this concern, we have made various changes across the entire paper detailing the choice of abstraction and how it helps model real world conditions.
> - It was hard to link the theorems with the text. To resolve this, we have added context to the theorem of where it fits in the entire framework more explicitly.
>
> For your ease, the changes are highlighted in blue.
> Before addressing the comments, we want to let the reviewer know that Section 6 is converted into a discussion section with conclusions and future work as a subsection. Subsection 4.3 (Extensions to Patrolling and Reachability) has been shifted to Section 6 as it might confuse the reader what are the contributions, and as such the entire Section is highlighted even though the major change is only reordering the content of the original submitted copy.
>
> ## Addressing the Weaknesses
>
> > The entire section 3, though clearly written by itself, is confusing to me as it is introduced before the problem formulation. I do not understand how the introduced technical tools like the non-reversible Markov chain and supervised pagerank relate to the problem we target to solve.
>
> We have now highlighted in related works that certain formulations do not consider the non-reversible nature of agent movement. It will help the reader with the relevance of the Non-reversible Markov chains for this work. In addition, the sampling process described in Subsection 3.1 (Non-Reversible Markov chain) is required to form the Sample Agent strategy that we describe in Subsection 5.1 (Agent Strategies).
>
> The algorithm we develop is a modified version of the Supervised Pagerank algorithm given by Bogolubskly et. al. (2016) which is mentioned in the second last paragraph of Subsection 4.2
>
> > Section 4, where the problem formulation is supposed to be presented, is also confusing to me. If I understand it correctly, the problem statement is hidden in L.224, which states that "The coverage problem can be defined as minimizing the expected information from all the $Y_t$ for all $0 \le t\le T$". This is not a rigorous mathematical statement and I still do not get what is the concrete problem we aim to solve from this sentence. I also do not understand the purposes of Lemma 1 and Theorem 1. I guess the properties revealed by Lemma 1 and Theorem 1 are crucial in terms of understanding the problem formulation, but I failed to see the connection here.
>
> We have modified the problem statement to “maximize the common information for random variables that represent node visibility” where node visibility is an indicator variable which is 1 if the node is visible from any agent for any time chosen uniformly at random. We define the probability associated with these indicator variables.
>
> Lemma 1 has been shifted to the Appendix. Theorem 1 is used to obtain a bound on the integral of $Y_t$ which then allows us to reduce the problem to minimizing the loss L. All this has been added so that it is clear to the reader what is the use of Theorem 1.
>
> > Section 4.3 is supposed to introduce the definition of Patrilling and Reachability problem. I can see that compared to coverage problem, the patrolling problem is to simply replace the set $V$ to $V_p$ in the definition of L. However, I do not understand the description about the reachability problem. It would be much clearer if putting the definitions of the three problems together at the beginning.
>
> As mentioned before, the extensions are shifted to discussion, as the goal of the subsection was to establish the flexibility of the formulation. The entire main body before the discussion talks about the coverage problem, and it is clearer to the reader what we want to convey.
>
> > All the lemmas and theorems are introduced abruptly. Though full proof details are provided in appendix, no proof sketch or high-level overview is provided in the main paper. This writing approach makes readers extremely difficult to grasp the significance and implications of each proof and the logical connections among the theorems and lemmas. Consequently, understanding the broader flow and purpose of the mathematical arguments are challenging.
>
> We have added explicit references to the Theorems throughout the text to ensure that the reader can link the theoretical results with the problem we are trying to address. While all proofs are deferred to the Appendix, the added context will be enough to understand the flow, as the proofs themselves do not contain anything that directly links with the main text.

---

> ### Author Response · Authors · 2024-11-16
>
> ## Addressing the Questions:
>
> > In section 5, it is mentioned that "We do experiments for the coverage problem and not for the rest of the extensions mentioned in subsection 4.3." Do you have any explanation or justification for this decision? Do the results on the coverage problem suggest any meaningful observation for the other two extensions?
>
> It was originally there to convey that the extensions are only to showcase flexibility. This is not fixed, as the subsection has been shifted to the discussion section. Another reason is that patrolling and reachability are heavily dependent on the agents' control strategy in comparison to coverage. Our work focuses on the planning aspect, and as such, it would be hard to do a fair evaluation.
>
> The coverage problem's results cannot be directly used to comment on what results we might expect from patrolling and reachability. For the same reason, the two problems themselves have a vast literature. However, the mathematical tools used for solving the other two are similar to those used for solving the coverage problem.
>
> > What is the rationale behind the 7 choices of the agent types in section 5.1? Are they commonly used in the literature?
>
> The following has been added to the paper to ensure that the reader understands why we select these strategies:
> “Our experiments model agents with unidirectional visibility and irreversible movement, which is a limitation for many previous works even though both conditions are found ubiquitously in real-world situations. Works that consider general agent movement and visibility become intractable for the size of environment and agent count that we consider. To our knowledge, no baseline provides a fair comparison for evaluation. Hence we select a few strategies that fit the task's requirements.”
>
> Selecting a baseline for fair evaluation is a significant issue given the wide variety and specific situations considered in various previous works. The main reasoning behind the selection of the strategies is as follows:
>  - Uniform random walk is the simplest stochastic strategy that can be used. Many times, simple things work in practice better than complex sampling strategies. Essentially, comparison to Occam’s Razor.
>  - Frontier-based exploration is a greedy approach that has been vastly successful in practice. Given that it is greedy, it is known to be not optimal. Comparing it with such methods highlights the performance difference between our approach and that of greedy approaches. This is important as in practice, many times, “a good enough” solution is more important than guarantees.
>  - The OptRandom strategy we introduce is not considered in the literature as it is not physically possible. However, it is similar to the “heavy hitters” adversaries used to evaluate coverage policies for intruder detection. The idea behind such evaluations is to give the agents a ridiculous amount of power to test against a proposed approach. As we mention in the paper, this is a strong baseline. While we do not know of any proof, intuitively, this is the best strategy in the case of uniform visibility. Defining uniform visibility is an arduous task, but it can be imagined as if the agent can sense anything within a circle of the agent.
>
>
> > The experiments are based on a synthetic environment. I'm wondering if this is a common benchmark and whether there are more realistic problem sets to test the performance of BOIL.
>
> To our knowledge, there are no common benchmarks, or standardized realistic problem sets to test BOIL's performance.

---

> > ### Comment · Reviewer_QVfK · 2024-12-02
> >
> > I appreciate the authors' response and it resolves most of my concerns. Considering other reviewers' opinions, I believe this work could benefit from a major revision to make the presentation much clearer.

---

### Meta-Review · Area_Chair_1NB5 · 2024-12-23

**Metareview:**

This paper presents BOIL (Blackbox Oracle Information Learning), a scalable approach for  navigating complex environments in multi-agent system using PageRank and common information maximization. The reviewers appreciate the following strengths:
- The paper considers a general problem and leverages different lines of works in their solution.
- Providing theoretical analysis to support the claim.
- Scalability advantages.

Weaknesses:
- Contribution and specific advances needs clarification.
- Relationship to existing work is not clearly discussed.
- Assumption of having access to an oracle needs further justification.
- Better connection to concrete application examples and comparison to approaches in specific domains are needed.

Overall, the reviewers share most concerns regarding presentation. The rebuttal and revision solved some issues. However, reviewers agree that  the submission would benefit from another round of revision to make the presentation clearer, with  three reviewers keep their ratings as 5. I agree with the reviewers and recommend rejection.

**Additional Comments On Reviewer Discussion:**

All reviewers shared major concerns about presentation issues in initial submission, including unclear contribution, insufficient discussion to existing works, assumption unjustified, lack of connecting to concrete application examples. Authors' rebuttal and revised version improved clarity but did not change reviewers' recommendation.

---

### Decision · Program_Chairs · 2025-01-22

Reject